# State of Knowledge on the Acquisition, Diversity, Interspecies Attribution and Spread of Antimicrobial Resistance between Humans, Animals and the Environment: A Systematic Review

**DOI:** 10.3390/antibiotics12010073

**Published:** 2022-12-31

**Authors:** Hélène Meier, Keira Spinner, Lisa Crump, Esther Kuenzli, Gertraud Schuepbach, Jakob Zinsstag

**Affiliations:** 1Institute of Veterinary Public Health, Vetsuisse Faculty, University of Berne, Schwarzenburgstr. 155, CH-3097 Liebefeld, Switzerland; 2University of Basel, Petersplatz 1, CH-4003 Basel, Switzerland; 3Swiss Tropical and Public Health Institute, Kreuzstr. 2, CH-4123 Allschwil, Switzerland; 4Department of Health Sciences and Technology, Eidgenössische Technische Hochschule (ETH) Zürich, Rämistrasse 101, CH-8092 Zurich, Switzerland

**Keywords:** anti-bacterial agent, drug resistance, bacterial, drug resistance, microbial, resistome, one health, health in social-ecological systems

## Abstract

Resistance to antibiotics is considered one of the most urgent global public health concerns. It has considerable impacts on health and the economy, being responsible for the failure to treat infectious diseases, higher morbidity and mortality rates, and rising health costs. In spite of the joint research efforts between different humans, animals and the environment, the key directions and dynamics of the spread of antimicrobial resistance (AMR) still remain unclear. The aim of this systematic review is to examine the current knowledge of AMR acquisition, diversity and the interspecies spread of disease between humans, animals and the environment. Using a systematic literature review, based on a One Health approach, we examined articles investigating AMR bacteria acquisition, diversity, and the interspecies spread between humans, animals and the environment. Water was the environmental sector most often represented. Samples were derived from 51 defined animal species and/or their products A large majority of studies investigated clinical samples of the human population. A large variety of 15 different bacteria genera in three phyla (Proteobacteria, Firmicutes and Actinobacteria) were investigated. The majority of the publications compared the prevalence of pheno- and/or genotypic antibiotic resistance within the different compartments. There is evidence for a certain host or compartment specificity, regarding the occurrence of ARGs/AMR bacteria. This could indicate the rather limited AMR spread between different compartments. Altogether, there remains a very fragmented and incomplete understanding of AMR acquisition, diversity, and the interspecies spread between humans, animals and the environment. Stringent One Health epidemiological study designs are necessary for elucidating the principal routes and dynamics of the spread of AMR bacteria between humans, animals and the environment. This knowledge is an important prerequisite to develop effective public health measures to tackle the alarming AMR situation.

## 1. Introduction

The O’Neill report reviewed the problem of antimicrobial resistance (AMR) in 2016, proposing recommendations from both economic and social perspectives. The landmark report estimated that, by 2050, ten million lives will be lost annually due to AMR [1]. In 2015, 671,000 infections and 33,000 deaths were associated with antibiotic resistant bacteria in the European Union alone [2,3]. Antimicrobial resistance genes (ARGs) can occur naturally [4], independent of antibiotic use, or develop under selection pressure during antibiotic treatment. AMR is acquired by the clonal spread of bacteria or by horizontal gene transfer (HGT); for instance, HGT can be performed by plasmids, which can also be termed “epidemic plasmids”, due to their ability of self-transmission [5]. The comprehensive genomic characterization of human pathogens shows that HGT of preexisting genes contributes to the majority of the current resistance problems generated by antibiotic use [6]. HGT of ARGs is shaped by phylogeny [7,8] and ecology [9], however, a comprehensive systemic understanding of the major drivers of ARG transmission and the direction of spread between the environment, animals and humans is still lacking [8,10]. Animals are a powerful mobile ARG pool that even potentially shape other resistomes (a collection of all ARGs of both pathogenic and non-pathogenic bacteria [11]) in other compartments [8,12]. Mobile ARGs mainly exist in four bacterial phyla: *Proteobacteria*, *Firmicutes*, *Bacteroidetes*, and *Actinobacteria* [8]. The tetracycline resistance genes, *tet (M)* and *tet (Q)*, and the integron-associated sulfonamide resistance gene, *sul1*, are the top three widely transferred mobile ARGs on the bacterial species level [8]. IncF plasmids are the most frequently described plasmid type of human and animal source [5].

In spite of the increasing prevalence of AMR bacteria-associated infections, the development of new antibiotics is currently stagnant. Although humans, animals and the environment are closely linked and interconnected in complex ways, the direction and dynamics of AMR spread between hosts, and commensal and pathogenic bacteria, still remain unclear [8,10,13]. Recommendations to reduce antibiotic use include the following: hygiene improvements, for instance, sanitary facilities, manure management, or wastewater treatment; better evaluation of indication for antibiotic usage in agriculture; and improved surveillance and rapid diagnosis in health care [1]. However, it is not known to what extent these recommendations would interrupt the emergence and transmission of AMR between humans, animals and the environment [10,13,14]. Information about how bacteria and their genes reach the environment and evolve over time is not enough to provide an understanding of their effects on human health outcomes [14]. Yet, the quantitative attribution of antibiotic resistance to specific transmission routes in different contexts is largely unknown [14]. Most AMR studies only assess point prevalence, are purely descriptive, investigate small sample sizes, and are limited in time and scale. Therefore, assessing resistance dynamics in complex interaction contexts, involving the three ‘compartments’ of the environment, animals and humans, is difficult [14].

In 2017 in Gothenburg, Sweden, the Joint Programming Initiative on Antimicrobial Resistance (www.jpiamr.eu (accessed on 15 August 2019)) stated the following critical knowledge gaps: How different sources contribute to antibiotics and antibiotic resistant bacteria in the environment, the environmental attribution (anthropogenic inputs in particular) on the evolution of resistance, the impacts of exposure to resistant bacteria from the environment on human and animal health, and the efficacy of technological, social, economic and behavioral interventions, to reduce environmental antibiotic resistance. Similarly, Woolhouse et al. [10] called for a sufficiently detailed, quantitative understanding of the dynamics of bacteria, drugs and resistance determinants in multiple host and environmental compartments to make meaningful decisions on AMR control [10]. Single discipline studies fail to identify the most effective methods of containing AMR. A lack of theory-driven One Health studies makes it impossible to identify effective mitigation strategies [15]. The Interagency Coordination Group on AMR recognizes that drivers of AMR reside within humans, animals, plants, food and the environment; thus, it is recommended that there are integrated One Health approaches for control, especially in low and middle income countries [16,17]. AMR is, therefore, not only a quintessential One Health issue [18], but One Health is an essential prerequisite for a systemic understanding of AMR acquisition and spread, and for identifying effective control strategies. It is appropriate to expand One Health to the broader concept of “Health in Social-Ecological Systems” (HSES), to include the environment as a third compartment alongside human and animal health [19,20].

There is still a significant lack of knowledge regarding the spread of antimicrobial resistance and the directions of spread, as well as effective causal pathways between hosts, and between commensal (microbiome) and pathogenic bacteria [8,10,21]. An in-depth understanding of HGT dynamics between the environment, animals and humans is essential to predict and prevent the emergence of antibiotic resistance [8], yet, very few studies assess AMR simultaneously in all three compartments. The few studies addressing this complex interaction have poor epidemiological planning [22] or assess only selected parts of the Social-Ecological Systems (SES) [23]. Only one known study used a comprehensive SES design combined with a metagenomic analysis. However, the authors sampled few animals within a cross-sectional study design, allowing for only limited statements concerning the attribution or spread of AMR [24]. The knowledge regarding behaviors, practices and contacts within SES is very scarce, despite the fact that these factors play a central role in understanding the acquisition and spread of AMR [15]. The acquisition and spread of AMR between humans, animals and the environment has not been investigated in detail, as very few longitudinal One Health studies integrating all these aspects are available. A comprehensive investigation requires sufficient statistical power, with representative sampling within human–livestock–environment contact networks. Moreover, the mapping of genes and bacteria, using the metagenomic and bioinformatic assessment of clonal diversity, is important to get a broader insight into shared ARGs and the common sources of acquisition [15]. Samples need to be collected over longer time periods, several months or even years, within a geographical range to cover all relevant host populations [10]. Bacteria are inextricably linked to the social life of their human and animal hosts, so interventions to improve public and animal health must be designed in response to dynamic food cultures and economies [25]. The complex interaction between humans, animals and the environment is an essential factor shaping the bacterial resistance landscape; therefore, it is important to investigate AMR dynamics using a One Health approach.

However, many studies attribute blame to certain compartments, usually without providing scientific evidence, instead of applying a holistic approach to investigate the spread of AMR. We hypothesized that there is no substantial evidence-based knowledge on the routes and directions of the spread of AMR between all three compartments; therefore, we undertook this review to summarize the current knowledge on acquisition, diversity and the interspecies spread of AMR between humans, animals and the environment.

## 2. Materials and Methods

### Literature Search

A detailed plan of the systematic review is provided in the Online Appendix A. This study is registered at the open science framework (DOI: 10.17605/OSF.IO/C5FUH). Briefly, the dataset, the search criteria and methods of criteria selection are described. The search was limited to publication dates from 2004 to 2019. We chose this time period because, since the year 2004, the number of publications on this topic has begun to increase significantly and, therefore, it was a realistic time range to investigate. A hermeneutic analysis, using Atlas.ti8^TM^ qualitative analysis software, assessed the investigated fields in humans, animals, the environment and the geographical location. Further, we examined the investigated bacteria and resistance genes, and the methods of AMR susceptibility testing. Attention was paid to studies on the direction of transmission and spread. Most of the studies investigated only one or two of the three compartments–either human and animal, human and environment or animal and environment compartments–but only few of them investigated all three compartments, in accordance with a HSES approach; this might give more insights into AMR dynamics. The study selection process can be seen in Figure 1. We included studies with the following criteria: Language–written in English, German, French, (Spanish and Italian were to be used but no papers were found in these languages); One Health approach–investigation of all three compartments (animal, environment, human); Topic criteria–investigation of AMR of bacteria to antibiotic agents; Format criteria–Paper provides new information (i.e., reviews, study protocols, and reports were excluded). Two reviewers screened all articles, discussing differences to find a consensus. We did not perform an analysis of risk bias because this study does not include a meta-analysis of risk factors; however, it did perform a qualitative assessment of the level of integration of studies at the human–animal–environment interfaces. We found 728 disagreements at the step “records considering 2 out of 3 compartments”. After further limitation to «considering all 3 compartments» and consensus discussion, 101 studies were excluded and no formal inter-rater agreement was performed, as shown in Figure 1 (PRISMA flow diagram).

Groupings of compartments were consensually made between the reviewers during the review process. They included the broader environmental compartment with highly specific sections, such as water or soil, but also included less specific sections, such as the closer animal environment. The specification of water sources was straightforward, as was the animal sector, for which most of the animal species could be identified. Foods sourced from animals were identifiable as from specific animal species, apart for a few specified exceptions. Human compartments were distinguished as of clinical or non-clinical origin, with some unknown sources. Bacterial genera and phyla could all be identified and were listed in the order of frequency of reporting.

## 3. Results

From an initial dataset of 9115 different publications that fit our search criteria (investigation of antimicrobial resistance in at least two of the three compartments as mentioned above), we finally included a total of 89 different publications into our systematic review (see Figure 1). The 89 publications analyzed include papers fulfilling the following criteria: papers that investigate all three compartments simultaneously (human, animal and environment); and only papers that provide primary data and an investigation of isolates, with regard to their resistance patterns and theoretical scenarios (e.g., epidemiological modeling).

### 3.1. Countries

Included publications cover 92 different countries (including oceans but excluding metagenomic datasets), with China (20%) and the United States (20%) most represented (see Figure 2).

### 3.2. Environmental Sectors

In total, 60 studies investigated water, in particular wastewater/sewage (Figure 3). Water was the environmental sector most often represented (67%) (detailed in Figure 4). In Figure 4 we distinguish between the different water sources in more detail. For overview and clarity purposes, we decided to only separate the environmental niches roughly, so as not to have too many categories. For example, other water sources include stagnant water, surface waters, water sediments, bathing water, and arctic snow.

### 3.3. Animal Species

Samples were derived from 51 defined animal species and/or their products (not considering animal samples/isolates of unknown species and metagenomic datasets; see Figure 5). Animal products (e.g., meat, chicken eggs, dairy products; see Figure 6), described in 41 studies (46%), were the most represented sector, followed by livestock birds, pigs and cattle.

### 3.4. Human Sector

The human sector consisted of samples (e.g., blood, feces, urine) from individuals of different ages, health status and country of origin. A large majority of studies (65%) investigated clinical samples of the human population (see Figure 7). Clinical samples are defined as samples coming from hospitals/in a medical context, e.g., blood or urine samples in clinics. Non-clinical samples were samples from people at home/community, and that did not have a hospital context.

### 3.5. Bacteria

A large variety of 15 different bacteria genera (see Figure 8) in three phyla (Proteobacteria, Firmicutes and Actinobacteria, in descending order; see Figure 9) (not considering unknown bacteria and metagenomic datasets) were investigated in the included studies.

### 3.6. Analysis

#### 3.6.1. Spread of Pheno- and Genotypic Antimicrobial Resistance

More than 95% of the 89 publications fulfilling the criteria for study inclusion, following the consensus, did not address the spread of AMR in detail. In Table 1, we summarize the overall findings. The majority (>80%) of the publications compared the prevalence of pheno- and/or genotypic antibiotic resistance within the different compartments. Based on these different comparisons, hypotheses concerning the spread of AMR were proposed: Busani and colleagues [26], for example, described a higher prevalence of resistance to single antibiotics linked to a higher frequency of multiresistant strains among certain compartments (comparing animal and food to humans); they drew the conclusion that farm animals and their food products might be the source of human infections with multidrug-resistant (MDR) bacteria [26]. However, scientific evidence supporting the hypotheses was often insufficient.

AMR spread between the three compartments occurs in different ways. Transmission can occur horizontally or vertically through ARGs, and it can also occur through AMR bacteria themselves exchanging between the different compartments. However, the latter form of transmission is controversial, as the exchange of AMR bacteria themselves between the different compartments does not necessarily lead to colonization of these “new” hosts, which is required for sustained conservation of ARGs. Furthermore, the expression of ARGs is dependent on extrinsic factors, as described below, and might not be conserved after an exchange into another compartment.

Scientific evidence for many hypotheses concerning AMR transmission dynamics between the different compartments is insufficient and often limited to descriptive or comparative studies, e.g., through comparison of the prevalence of AMR bacteria and/or ARGs in different compartments. A higher prevalence of AMR bacteria/ARGs in certain compartments is often attributed to the emergence of this specific antibiotic resistance within the compartment. Furthermore, the detection of similar or identical resistance genes within different compartments is often considered as proof of AMR transmission. However, the latter scenario could also occur if the involved compartments are subjected to the same or similar selection pressure and/or previously unknown driver. The investigation of mobile genetic elements, as well as their transmission abilities, e.g., filter matings for the investigation of plasmid transmission [27,28,29], likely provide more solid evidence for assessing transmission dynamics within different compartments; however, these studies are, to date, limited to in vitro systems. More detailed phylogenetic studies describing specific ARG mutations, as well as the distribution and prevalence of different ARGs, could provide more evidence for the hypotheses concerning transmission dynamics. However, many studies do not provide an integrated analysis of ARG prevalence and the phylogenetic situation to assess the similarity of different isolates within the same or different compartments [30]. There is evidence for a certain host or compartment specificity regarding the occurrence of ARGs/AMR bacteria [29,31,32,33,34]. This could be compatible with the concept of the very limited AMR spread between different compartments, for example, only in the context of very close contact, such as people living in close proximity to animals (e.g., farming communities), or occupationally exposed people, such as farmers or veterinarians exposed to pigs [35].

Gatica and colleagues [33] reported a close phylogenetic relationship between ARGs (beta-lactamase) found in bovine and human feces forming clusters with relatively high levels of connectivity; this suggests the spread of AMR between these two sectors, although the direction of spread remains unknown. However, in their metagenomic study, the authors reported a generally low similarity between environmental clusters, compared to human and bovine feces pulsed-field gel electrophoresis (PFGE) clusters. Together, these results suggest a specificity of ARGs to particular environments and that ARG transfer between different environments is not very common [33], as also reported by Kyselkova et al. [36]. Gatica and colleagues [33] argue that “The significant differences in bacterial community composition between the ‘fecal’ and ‘environmental’ samples suggest that horizontal transfer of ARGs is often constrained by phylogenetic boundaries as previously indicated for the soil microbiome [7]. Furthermore, other factors such as founder effects, ecological connectivity between donor and acceptor, fitness costs or second-order selection may limit the gene transfer of ARGs between environments [7,37,38] […] supporting the hypothesis that natural environments may be reservoirs of precursor ARGs that can eventually emerge in human pathogens [39]. Conversely, these low levels of identity also indicate that pathogen-associated ß -lactamases are not profuse in natural environments. Albeit the resistome hypothesis [40], the low connectivity observed between natural ß-lactamase clusters and fecal clusters in the network analysis, suggest that these events occur at relatively low frequencies.” [33] The authors suggest further analysis and studies concerning the spread of disease between compartments [33].

#### 3.6.2. Antimicrobial Resistance as a Multifactorial Problem

Different publications [32,41,42,43] have stated that the emergence of antibiotic resistance might not only be linked to the usage of antibiotics in the healthcare sector. Nesme and colleagues [42] described vancomycin-resistant genes as being the most abundant antibiotic-resistant gene determinants-annotated reads in metagenomic studies, using human datasets. However, vancomycin use is rare as it is highly regulated and limited to specific indications as a last resort antibiotic treatment. In contrast, the prescription of amoxicillin in clinical medicine is much more common; nevertheless, amoxicillin-related resistant genes, such as beta-lactamases, are less abundant in the metagenome of human stool samples [42]. Together, these findings suggest that many more factors, beyond the prescription frequency of antibiotics in clinical medicine, influence the emergence and spread of antibiotic resistance. Similarly, another study investigating multidrug-resistance in different Listeria monocytogene strains [44] found no resistance to beta-lactam antibiotics (ampicillin) and aminoglycosides, despite these antibiotics being the clinical treatment of choice. However, the authors could detect resistance to other antibiotics in 145 out of 259 Listeria strains.

Numerous publications [27,31,35,45,46,47,48,49,50,51] have described other factors having an impact on the distribution, prevalence and transmission of AMR bacteria/ARGs; these include the time frame and geographic location of sample acquisition [31,46], various environmental factors (e.g., soil composition might influence the persistence and transmission of AMR bacteria [51]), the season at the time of sample collection [47,51], and the age distribution within the study population [49]. Furthermore, inter-individual genetic variants might have an impact on study results [27,35,45]. This highlights the variety of factors that potentially influence AMR spread, dynamics and distribution; this makes it hard to understand AMR in its entirety and makes AMR control even more difficult. Additionally, this complicates study design, since all of these different factors need to be considered during the planning of field studies, in order to minimize any bias.

The high need for the better understanding of such factors, as detailed below, is best addressed using a One Health/HSES approach.

Seasonality: According to Hu and colleagues [47] no significant seasonal variations, in terms of mean antibiotic resistance frequency, could be observed in river water samples. However, obvious variations in the antibiotic resistance spectrum were observed, with the frequencies of three- to five-drug resistance increasing in winter, compared to summer; conversely, the mean multi-antibiotic resistance index was lower in summer than in winter. The authors attributed this phenomenon to a higher persistence or discharge rate of multi-resistant E. coli in winter. Ma and colleagues [50] confirmed this seasonal variability, observing two different peaks in human Salmonella enteritidis isolates; here, the peak value of isolation of MDR S. enteritidis is in spring (May), being temporally shifted from the peak value of isolation of S. enteritidis in humans, in general, in summer (August) [50]. The mechanisms leading to this seasonality remain to be elucidated. One could hypothesize that either the therapeutic regime (i.e., choice of antibiotic agent) or seasonal variation in exposure to (multidrug-resistant) bacteria (e.g., higher prevalence in the general population in a specific season due to consumption of specific food, such as meat, during barbecue season, or leisure activities, such as swimming) led to the two different peaks in spring and summer.

Time frame and geographical location: Different studies [31,46] described that the prevalence of antibiotic resistance is highly dependent on the geographic location and time of the sample acquisition. Accordingly, it is crucial to clearly state the time and location of the sample acquisition, and/or limit research to a specific geographical location, in order to reduce the spatio-temporal bias and decrease the undesired exogenous variability. Several studies described a highly variable prevalence of antibiotic resistance over time, with either increasing or decreasing trends [34,52,53,54]. Accordingly, the comparison of samples acquired at different time points can be very difficult. Therefore, it is essential to conduct studies on temporally and spatially related isolates (within, as well as between, different compartments) to minimize unspecific variation and bias. Nevertheless, several studies analyzed and compared samples between and/or within compartments that were not time [35,55,56] and place matched [57,58]. These factors should also be considered when analyzing and interpreting long-term studies conducted over a time frame of several years [28,43,44].

Sampling site within the same compartment: Ibekwe and colleagues [48] demonstrate that even samples taken within the same compartment and geographical sampling site showed differences in AMR patterns. According to PFGE fingerprinting, *E. coli* isolates were more diverse in surface water than in sediment, with more complex AMR profiles found in isolates from surface water than from sediments. The authors suggested that sediment host resident bacteria populations, whereas more transient populations are present in surface water [48].

Age categories: Mutters and colleagues [35] describe different abundances of nasal methicillin-resistant *Staphylococcus aureus* (MRSA) carriage, depending on the age and production category of pigs; here, porkers showed the highest rate, followed by piglets and sows. Ma and colleagues [49] also showed an age-dependent variation in AMR abundance, with the presence of ARGs being significantly enriched in the feces of adult chickens, but decreased with the increase in age in pigs. One possible explanation could be the application of antibiotic agents; antibiotics are administered to chickens during the whole production period, whereas they are administered on a decreasing basis during pig growth. Of course, the setting and type of production system, as well as the (time point) administration of antibiotics, vary between different countries.

Genetic variation: It is hypothesized that a limited gene pool with low inter-individual variability, as is present, for example, in the chicken industry, where there are only a few genetic lines of primary breeding flocks utilized, might be associated with a lower ARG-gene pool and lower resistome variability, respectively [27,35,45].

Atterby and colleagues [27] investigated all three compartments and different animal species (Swedish gulls, poultry, pigs, calves). While other animal species harbored different extended-spectrum beta-lactamase (ESBL) genes, only one type of gene could be found in ESBL-producing *E. coli* (blaCTX-M-1) isolates from poultry. Antunes and colleagues [45] report similar findings: The different characteristics of animal production in poultry and pork seem to be in concordance with the prevalence and diversity of integron types, as well as the number of clones in these two species, being higher in pork than poultry products. A difference in the antibiotic selective pressure after years of intensive use could contribute to the spread of specific, perfectly host-adapted integrons and clones.

Mutters and colleagues [35] observed the same phenomenon in a very homogeneous population of AMR bacteria in pigs, where livestock-associated MRSA belong exclusively to one clonal complex (CC398), in contrast to hospital-associated MRSA with a wide variety of different MRSA strains. Mutters [35] posited the same reasons as Atterby [27] for his findings, possibly also having an impact on other compartments: human beings with a limited gene pool (e.g., present in small communities with low genetic variability) are more prone to sharing similar ARGs. This could be an important confounder when investigating geographical differences in ARG prevalence. Similarly, such genetic alterations within specific populations could also act as a barrier, preventing the transmission of some ARGs/AMR. The mechanism remains to be elucidated and has not been investigated in further detail. One hypothesis is that certain genetic patterns or profiles might promote or prevent the colonization of specific AMR bacteria. This factor could influence study outcomes and limit the extrapolation of study results to other populations or geographical regions.

Fitzpatrick and Walsh [59] attributed an additional role specifically to the gut microbiome, which shows a small variability in humans; this, therefore, contributes to a less variable abundance of ARGs in humans [59] and human-impacted environments [33].

**Table 1 antibiotics-12-00073-t001:** Synopsis of findings on existing integrated studies on AMR out of a total of 89 studies.

Factor Examined	Key Finding	Comment
Number of studies	Only about 1 percent of studies consider two or more compartments	Out of 9115 studies
Countries	More than 20% of the studies each, were conducted in China and the United States of America	Studies were conducted in 92 countries.
Environmental sectors	Most studies were on water (67%), soil (26%), food (20%) and close animal environment (18%)	Other sectors were human hospital environment, close human environment, plant, sludge, sea food and industry
Water	Wastewater was most often examined (31% of the studies)	Other water sources were sea, lake, pond and river water
Animals	Cattle (29% of the studies) and pigs (36%) were the most important individual species	Other animals studied were sheep, goat, dogs, cats, rabbits, insects, rodents, horses, fish and wild birds
Meat	Chicken (33%) and pork meat (25%) was most often examined	Other meat was from turkey, dugs, sheep, fish, eggs and milk
Human samples	Human samples were mostly clinical samples (65%)	
Bacterial species	Salmonella (26%), Escherichia coli (19%) and Enterococci (9%) where the most important identified bacteria	Others were Listeria, Aeromonas, Clostridium, Campylobacter and Pseudomonas.
Bacterial phyla	Proteobacteria (66%) were the most studied phyla	Other phyla were Firmicutes and Actinobacteria

## 4. Discussion

Inadequate prescribing and the overuse of antibiotics in livestock production and agriculture are considered the most important factors behind the emergence of antibiotic resistance, but numerous studies demonstrate that AMR is much more complex and many additional factors influence its presence and spread [27,31,35,45,46,47,48,49,50,51]. Resistant bacteria are disseminated through both ground and surface water [1,21]. In some low and middle income countries, bacteria with AMR are even detected in drinking water [60]. It is well known that antibiotic resistance is steadily increasing and is associated with the increasing burden of disease worldwide. The monitoring of antibiotic resistance is an important parameter in assessing the distribution and prevalence of different resistant bacteria strains. However, in order to further contain the spread of antibiotic resistance, it is essential to investigate the dynamics and possible transmission of AMR bacteria and/or resistance genes using a broader multi-systematic approach. This could allow the further investigation of relevant questions: How does the spread of resistance occur? Does transmission occur within only one compartment or occur mainly between different compartments? What is the relevance of mobile genetic elements regarding AMR transmission? Are there common sources of antibiotic resistance acquisition for different compartments? So far, these questions have not investigated in detail, with most studies limited to the description of AMR prevalence [54]. Data collection is the first step to understand the process of antibiotic resistance emergence and transmission. However, this data must also be integrated into a systematic approach by further elaborating upon epidemiological analysis, such as with the concept of HSES. The investigation of genetic changes, using metagenomic sequencing [33,49] and whole genome sequencing [34], could be an approach to better consider all the factors involved in AMR transmission. Stringent One Health/HSES epidemiological study designs, using cohort or case-control designs, minimize the potential sources of bias; therefore, they are necessary for elucidating the principal routes and dynamics of the spread of antibiotic resistant bacteria and antibiotic resistance genes between human, animal and environmental compartments.

It is striking that many studies are focused on the responsibility of certain compartments for the current AMR problem, instead of providing more evidence-based data. Many studies stated that the high usage of antibiotics in veterinary medicine is one of the main drivers of emerging antibiotic resistance [44], and resistance could then be transmitted to human beings through food [44,45,61]. However, evidence for this statement is scarce, being limited to the high prevalence of AMR in animals; this could also be due to the human or environmental cross-contamination of animal products [61]. Furthermore, most studies did not consider or investigate the direction of spread between compartments, with a few exceptions [35]; most frequently, they suggest the transmission of antibiotic resistance from humans to animals [31,45,55,61,62]. There is a high need for epidemiological studies with source tracing and whole genome sequencing, allowing for phylogenetic analyses.

Limitations of the study methodology: The primary objective of this study was a qualitative systems analysis. We did not perform a formal analysis concerning the risk of bias in the included studies because we performed no quantitative meta-analysis of the data. We limited our search to one database (Pubmed). The comparability between different countries may be difficult as antibiotic resistance dynamics can vary between different countries, as well as between the antibiotics used/prescribed. Limitations may also include risks of bias, such as selection bias, attrition bias, publication bias, inadequate blinding, selective outcome reporting, and inconsistencies that may lead to clinical and statistical heterogeneity.

### Towards Better Integrated Study Design

Most of the limitations encountered in the publications analyzed for this review involved the study design (specifically insufficient epidemiological planning). In many studies, isolates are poorly described [22,26], with insufficient or the absent documentation of the sampling location, time and species. As mentioned above, these factors might be crucial and significantly affect study outcomes. Furthermore, some studies compare isolates sampled in different geographical regions [57,58] or at different time points [35,55,56], which might lead to systematic spatial biases.

Another problem involved the separation of the different compartments. Some studies [44,54,63] investigated the three different compartments and then pooled them for downstream analysis, such as AMR prevalence calculations. With this method, no conclusions about transmission between different compartments can be drawn.

The limited sample size [22,30,53,64,65] and uneven distribution of samples in different groups, for instance between different compartments and/or species, can significantly affect study outcomes. In several studies, very few environmental samples were compared to a much larger number of human isolates [50,55,66]. When analyzing AMR prevalence between different compartments or analyzing clusters in PFGE, the uneven distribution of samples within different groups can lead to a significant sampling bias; these include clusters, being driven uniquely by uneven sample distribution [50,63]. This limitation could easily be overcome through better epidemiological study planning.

Another bias seen in the publications included in this review involved the populations studied. Most of the studies including animal samples, investigated animal species in terms of their high relevance for human beings, such as pets or livestock and their products. Although this allows statements about transmission dynamics between different compartments that are in close relation, this could lead to a spatial bias; this is because AMR emergence in each compartment is no longer independent of the other compartments due to their closely linked habitats.

Most of the studies are limited to a subset of resistance genes [33,67] or phenotypical resistance [68], introducing a selection bias due to only testing selected antibiotics. In cases where studies investigate only specific antibiotic resistance genes, rather than using an unbiased study approach, one might miss the presence of other or concomitant significant resistance genes. Moreover, the presence of ARGs does not necessarily translate into phenotypical resistance and vice versa [48]. This can be due to, for example, previously unknown ARGs or the lack of ARG expression because of mutations, such as inactivating deletions [57,61,69]. If studies investigate both phenotypical and genotypical resistance simultaneously, they can overcome this limitation. Some studies [53] assumed that the occurrence of similar or identical PFGE patterns and antibiotic resistance phenotypes between different compartments might be evidence for the spread of AMR. The investigation of the underlying genotypic antibiotic resistance of the isolates would have been more informative, since other reasons could also lead to such findings, such as compartments being exposed to the same selection factor or the same source of ARGs; this, therefore, leads to identical resistance phenotypes.

Despite the strict inclusion criteria chosen for this systematic review (investigation of dynamics of antibiotic resistance in all three compartments including humans, animals and environment), many of the remaining publications only assessed the prevalence of phenotypical and/or genotypical resistance, without an integration of epidemiological analyses [54,68,70]. Accordingly, statements about potential transmission routes and the dynamics of bacteria displaying antibiotic resistance, were often omitted or very limited. In many publications, hypotheses were posited; however, the epidemiological analyses required for experimental validation, such as source tracking or phylogeny, were often scarce or lacking.

Furthermore, the approaches chosen for the investigation of source tracking/phylogeny analysis, were often of insufficient experimental quality. Many studies used the PFGE, however, the authors applied this method using only a single restriction enzyme [50,71] for the comparison of the genetic relatedness of AMR bacteria of different origins. It has been reported by different groups [72,73,74] that it is highly important to use a second restriction enzyme to confirm the phylogenetic relationship between isolates through more detailed and exact macro-restriction profiling/clustering. This finding was supported by Keelara and Thakur [29]. Restriction analysis of plasmids, from all three compartments with only one enzyme (EcoRI), exhibited a similar restriction profile, suggesting the presence of the same plasmid. However, when these isolates were further analyzed with two additional restriction enzymes (HindIII and PstI), environmental and animal plasmids still exhibited a similar pattern, while the human isolate showed a different restriction profile; this allowed a better distinction of the phylogenetic relationship.

In the studies included in this review, the One Health principle was often overlooked when analyzing and discussing the data. Compartments and interfaces between them were not considered all together; often only two were compared, while the third one was disregarded [41,47].

Best practices for integrated One Health studies on AMR should thus include the following minimal standards: (a) State a clear hypothesis of the study; use standard epidemiological study design (cross-sectional, cohort, case-control) with statistical power calculation depending on the research question; (b) Comprehensive reporting of spatio-temporal metadata, that allows the relation of causal pathways between the different compartments (humans, animals, food, environment); (c) Clarify the study objective for specific resistance genes or the whole resistome. Relate phenotypic and genotypic resistance; (d) Attempt an assessment of an incremental benefit of knowledge, cost and disease burden from an integrated study design [19].

## 5. Conclusions

There is evidence that antibiotic resistance is closely linked to specific hosts and/or compartments [27,45], which could be due to their limited gene pool. However, this host-specificity could also be linked to very close contact between individuals within a specific sector (e.g., poultry flock), leading to a shared bacterial population and shared AMR. Moreover, it has been shown that very close contact between individuals of two different compartments, such as pigs and farmers [35], can also lead to resistance transmission, overcoming the suggested host-specificity.

Only very few studies investigated the environmental compartment in detail when looking at the spread and transmission dynamics of antibiotic resistance. However, the environment might be an important component of this system, especially considering it as an interface between humans and animals. This further enforces the need for expanding the One Health to a HSES concept, allowing a better investigation of all influencing factors and their complex interplay; this should be supported by various studies [27,31,33,34,52,55,59,75,76].

A similar systematic review has been performed for integrated studies on antimicrobial resistance genes in Africa and provides guidance for prospective integrated cohort studies [77]. A further review should outline the key drivers of antibiotic resistance in humans using a One Health approach [78]. These studies additionally contribute towards best practices for integrated One Health studies on AMR.

## Figures and Tables

**Figure 1 antibiotics-12-00073-f001:**
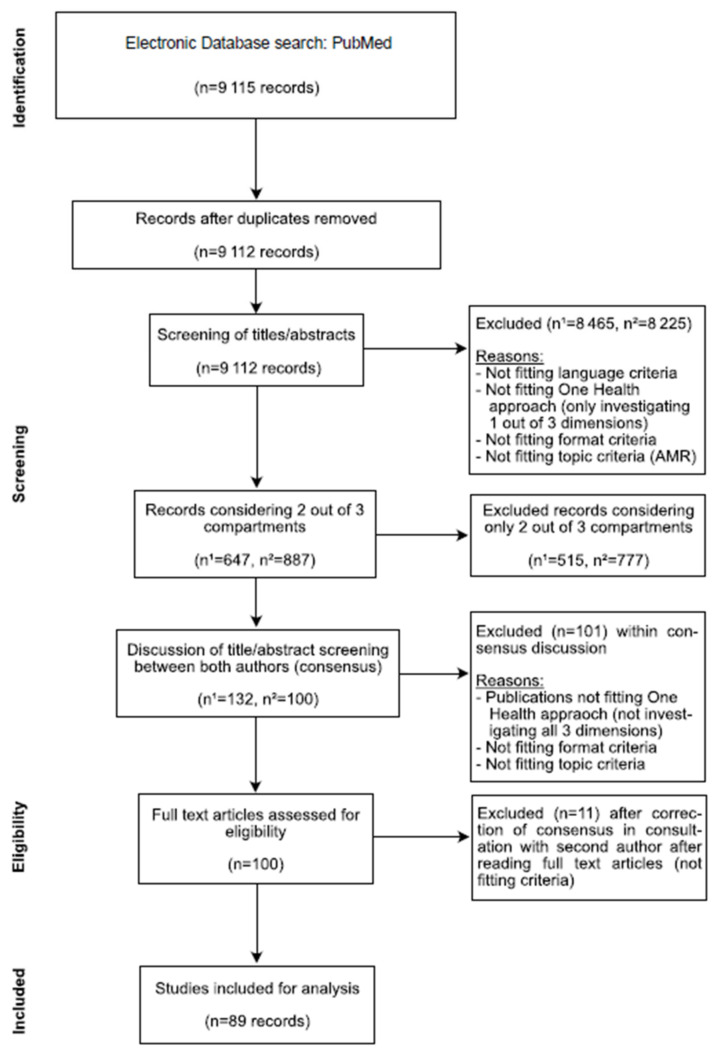
PRISMA flow diagram of study selection (n^1^: H. Meier; n^2^: K. Spinner).

**Figure 2 antibiotics-12-00073-f002:**
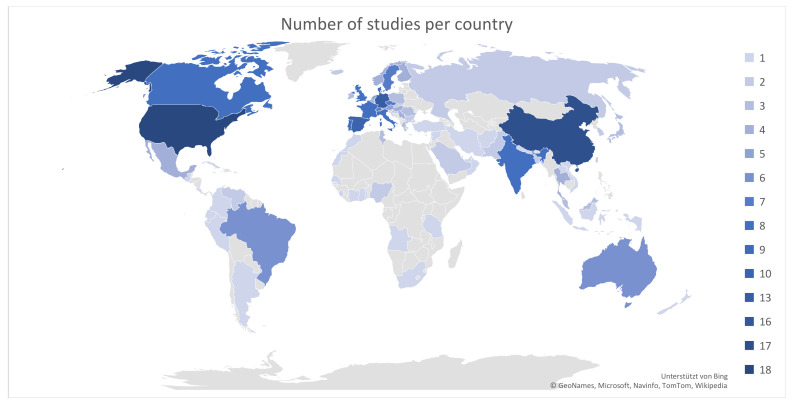
Number of studies per country (color coded). Grey color means that no studies that fit inclusion criteria were available.

**Figure 3 antibiotics-12-00073-f003:**
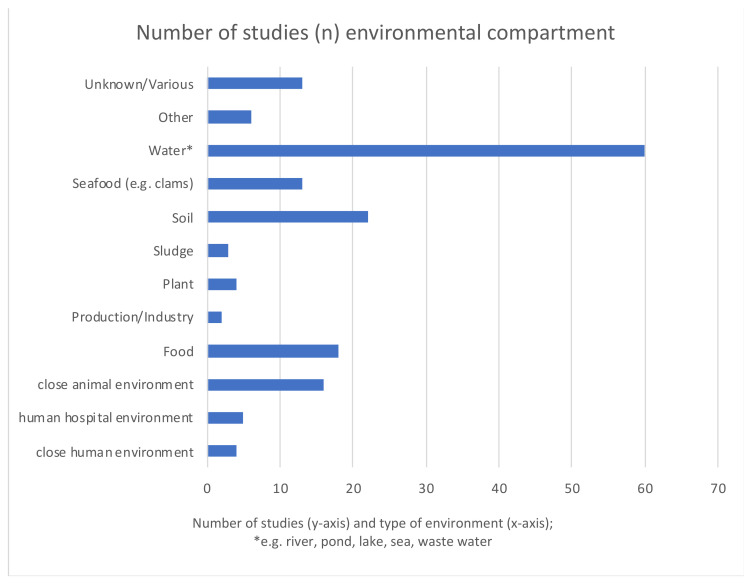
Environmental niches represented in selected studies. “Close animal environment” refers to the holding place of animals. “Close human environment” refers to the household where humans live.

**Figure 4 antibiotics-12-00073-f004:**
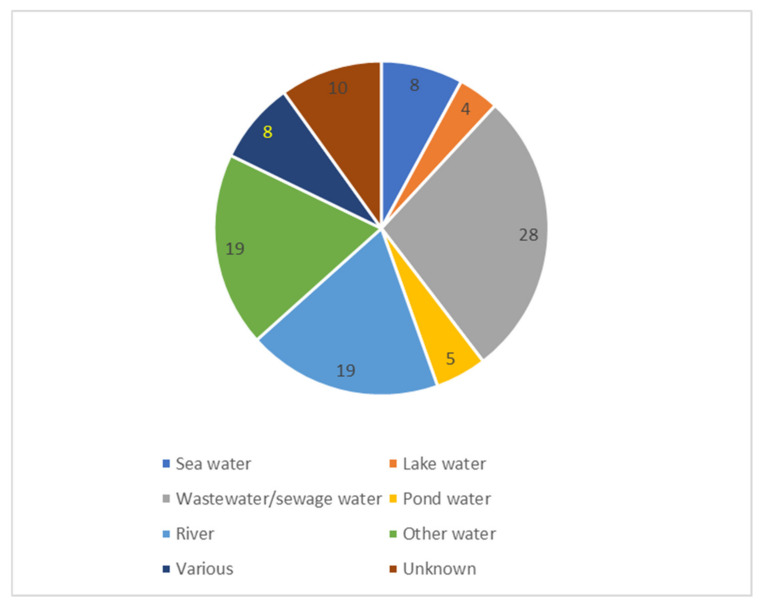
Type of water sources represented by number of included studies (*n* = 101). “Other water” are other sources than those specified. “Various” sources refer to more than one of those uniquely specified.

**Figure 5 antibiotics-12-00073-f005:**
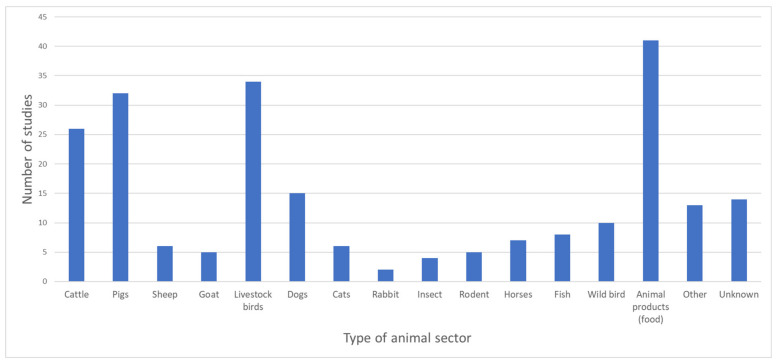
Number of studies (*y* axis) per type of animal sectors (*x* axis) represented in all 89 studies included. “Animal products” comprised the following: Meat (chicken, turkey, duck, sheep, cattle/beef, fish, pork, offal, not specified), chicken eggs, dairy products (cheese, milk, not specified), and unknown/various. “Livestock birds” comprised the following: Chicken, duck, turkey, goose, ostriches, and other (Hoatzin Cecum). “Fish” compromised the following: Rutilus rutilus, wild eels, salmon, and freshwater fish. “Other” comprised the following: Frog, monkeys, wild boars, peacock, parrot, snake, buffaloes, turtles, red fox, dinoflagellate, dolphin, whales, wolf, other domestic birds, shells, and “other reptiles”.

**Figure 6 antibiotics-12-00073-f006:**
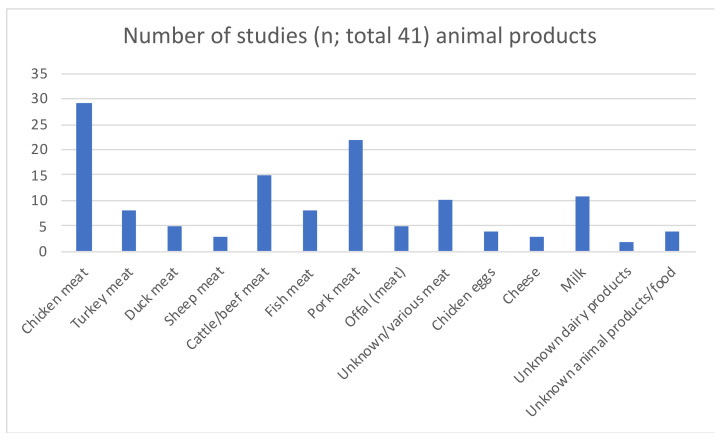
Type of animal product/food of animal origin represented by included studies.

**Figure 7 antibiotics-12-00073-f007:**
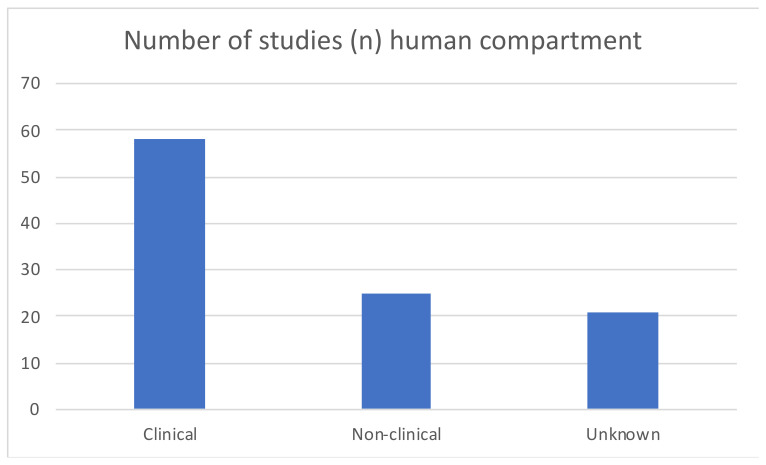
Human compartment sample types represented by included studies.

**Figure 8 antibiotics-12-00073-f008:**
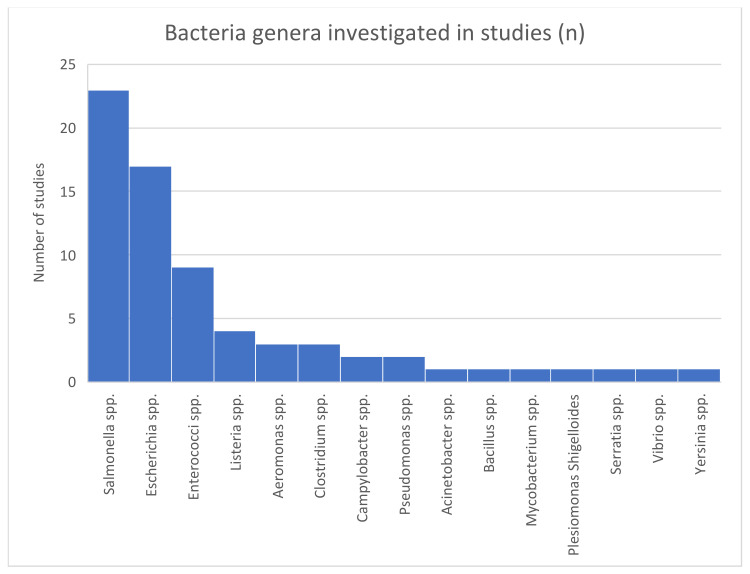
Bacteria genera investigated by included studies.

**Figure 9 antibiotics-12-00073-f009:**
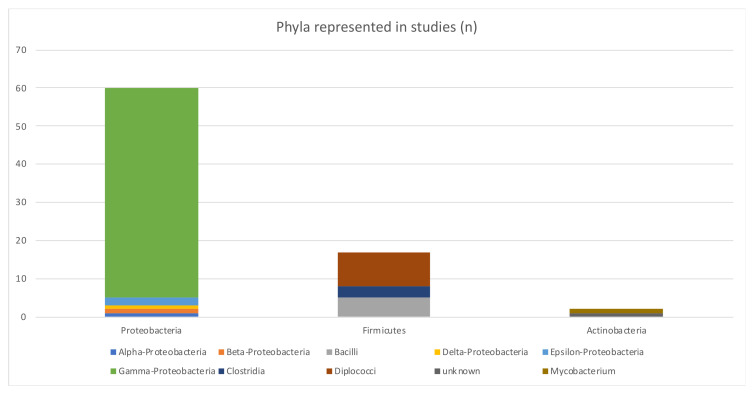
Bacteria phyla represented by included studies.

## Data Availability

The data is available in the Appendix A.

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
