# Peer review of "State of Knowledge on the Acquisition, Diversity, Interspecies Attribution and Spread of Antimicrobial Resistance between Humans, Animals and the Environment: A Systematic Review"

_antibiotics, 2022, doi:10.3390/antibiotics12010073_

Round 1

Reviewer 1 Report

In my opinion, the work presented for evaluation is very innovative because it contains a review of the literature in the field: State of knowledge on the acquisition, diversity, interspecies attribution, and spread of antimicrobial resistance between humans, animals, and the environment. This topic is crucial as it constitutes a serious clinical challenge for health policy in the era of the overuse of antibiotic therapy in everyday medical practice.
Below are some of my suggestions to improve the scientific value of the manuscript:
1. Abstract- no refers to the obtained results and conclusions of the literature review, but mainly limits to the importance of the literature review (the reader is hardly ever to read the full text of the manuscript).
2. To better understand the manuscript, I propose the following chronology of individual parts of the manuscript: 1. Introduction 2. Material and methods 3. Results 4. Discussion (at the end of this section should be a subsection: study limitations). 5. Conclusions
3. The methodology lacks information about the time horizon of the systematic review; moreover, was the inclusion criterion for the review the year of publication - the study no older than 5-10 years? The review included studies published in 2004-2019, and what about the latest literature in 2020-2022 (was this type of research conducted in the era of the Covid-19 pandemic?)
4. Line 49 What does the phrase in reference [8] "Error! Bookmark not defines" mean?
5. Line 135 sentence begins with a lowercase letter.
6. Figure 4. There are no numerical values or percentages in the pie chart.
7. Figure 5 is not legible. I suggest using a larger font.
8. throughout the work, there is no space after the last word in the sentence and a reference.

Author Response

Dear Editor,

Many thanks for the reviews and the opportunity to submit a revised version of the manuscript. Below please find a letter of revision with our answers preceeded by the word Answer.

We hope the revision meets your expectations at remain at your full disposition for further work if needed.

With best regards

Jakob Zinsstag

Reviewer 1

In my opinion, the work presented for evaluation is very innovative because it contains a review of the literature in the field: State of knowledge on the acquisition, diversity, interspecies attribution, and spread of antimicrobial resistance between humans, animals, and the environment. This topic is crucial as it constitutes a serious clinical challenge for health policy in the era of the overuse of antibiotic therapy in everyday medical practice.
Below are some of my suggestions to improve the scientific value of the manuscript:

Answer: We thank the reviewer for the encouragement.

  1. Abstract- no refers to the obtained results and conclusions of the literature review, but mainly limits to the importance of the literature review (the reader is hardly ever to read the full text of the manuscript).

Answer: We added obtained results and conclusions to abstract

  1. To better understand the manuscript, I propose the following chronology of individual parts of the manuscript: 1. Introduction 2. Material and methods 3. Results 4. Discussion (at the end of this section should be a subsection: study limitations). 5. Conclusions

Answer: We structured the paper according to the reviewer’s request.

  1. The methodology lacks information about the time horizon of the systematic review; moreover, was the inclusion criterion for the review the year of publication - the study no older than 5-10 years? The review included studies published in 2004-2019, and what about the latest literature in 2020-2022 (was this type of research conducted in the era of the Covid-19 pandemic?)

Answer: We agree with the reviewer and give the time period of the analysis. We add the sentences: “The search was limited to publication dates from 2004 to 2019. We have chosen this time period because since the year 2004 the number of publications on this topic began to increase significantly and it was a realistic time range to investigate.” As the Covid-19 pandemic emerged in 2020, there was a substantial impact on the social-ecological systems and we propose to keep this review at a pre-pandemic state to avoid confusion with different overlaying processes.

  1. Line 49 What does the phrase in reference [8] "Error! Bookmark not defines" mean?

Answer: The phrase is removed

  1. Line 135 sentence begins with a lowercase letter.

Answer: The sentence is corrected.

  1. Figure 4. There are no numerical values or percentages in the pie chart.

Answer: the Figure has been improved.

  1. Figure 5 is not legible. I suggest using a larger font.

Answer: the Figure has been improved.

  1. throughout the work, there is no space after the last word in the sentence and a reference.

Answer: Spaces have been included throughout the manuscript.

Reviewer 2 Report

This review is comprehensive and the selection criteria for inclusion of source references is thoroughly documented. It is an important topic and the authors rightly emphasise the need for a holistic –rather than a compartmentalized - approach in studies focusing on AMR.

The presentation of some of the information requires further work:

The abstract needs re-working as it is highly repetitive in places. For example, lines 16-17: This sentence requires correction, as the meaning is unclear and “remain” appears twice in short succession.

Line 135: there appears to be some text missing here.

Figure numbers need to be fixed, as the first figure presented is labeled as Fig 2 and the final one as Fig 1.

Figures incorporating graphs need to be modified to include x and y axis titles, with all other information  (including number of studies included) placed in the figure legend.

Author Response

This review is comprehensive and the selection criteria for inclusion of source references is thoroughly documented. It is an important topic and the authors rightly emphasise the need for a holistic –rather than a compartmentalized - approach in studies focusing on AMR.

Answer: We thank the reviewer for his encouragement.

The presentation of some of the information requires further work:

The abstract needs re-working as it is highly repetitive in places. For example, lines 16-17: This sentence requires correction, as the meaning is unclear and “remain” appears twice in short succession.

Answer: The abstract has been reworked and the word remain has been removed.

Line 135: there appears to be some text missing here.

Answer: The sentence has been corrected.

Figure numbers need to be fixed, as the first figure presented is labeled as Fig 2 and the final one as Fig 1.

Answer: Figure numbers have been fixed.

Figures incorporating graphs need to be modified to include x and y axis titles, with all other information  (including number of studies included) placed in the figure legend.

Answer: Figures have been improved accordingly.